# Treated Wastewater Irrigation—A Review

**Mahmoud S. Hashem [1,2] and Xuebin Qi [1,\*]**

1   Farmland Irrigation Research Institute, Chinese Academy of Agricultural Sciences, Xinxiang 453003, China; mahmoudsabry375@gmail.com
2   Agricultural Research Center, Agricultural Engineering Research Institute (AEnRI), Giza 256, Egypt
\*   Correspondence: qxb6301@sina.cn

**Abstract:** As the most important resource for life, water has been a central issue on the international agenda for several decades. Yet, the world's supply of clean freshwater is steadily decreasing due to extensive agricultural demand for irrigated lands. Therefore, water resources should be used with greater efficiency, and the use of non-traditional water resources, such as Treated Wastewater (TW), should be increased. Reusing TW could be an alternative option to increase water resources. Thus, many countries have decided to turn wastewater into an irrigation resource to help meet urban demand and address water shortages. However, because of the nature of that water, there are potential problems associated with its use in irrigation. Some of the major concerns are health hazards, salinity build-up, and toxicity hazards. The objectives of this comprehensive literature review are to illuminate the importance of using TW in irrigation as an alternative freshwater source and to assess the effects of its use on soil fertility and other soil properties, plants, and public health. The literature review reveals that TW reuse has become part of the extension program for boosting water resource utilization. However, the uncontrolled application of such waters has many unfavorable effects on both soils and plants, especially in the long-term. To reduce these unfavorable effects when using TW in irrigation, proper guidelines for wastewater reuse and management should be followed to limit negative effects significantly.

**Keywords:** treated wastewater; irrigation; soil properties; macronutrients; crop quality; heavy metals

## 1. Introduction

Water is an extremely important natural resource because life cannot exist, and industry cannot operate, without it. Water plays an essential role in the growth of countries because a steady supply of fresh water is a crucial prerequisite for establishing a permanent community. Yet, the world's supply of clean freshwater is steadily decreasing. In many countries, the water demand surpasses the supply, and as the world population continues to rise and demands for water increase, freshwater shortages have emerged [1–3]. Irrigation is considered the main user of freshwater. Irrigation of land accounts for approximately 80% of the total freshwater usage [3], and it will account for an additional 15% by 2030 [4,5], which will cause water crises in the regions that suffer from water shortages, such as the Middle East and North Africa region, for example. In addition, [6] mentioned that by 2025, nearly 1.8 billion people will live in a region that suffers from water shortages. Therefore, it is essential to use alternative sources of water. Within the following decades, it is estimated that over 40% of the total population will confront water stress or scarcity, representing a meaningful impact on water security [7]. Therefore, the reuse of wastewater is an important asset for agricultural purposes [8]. Wastewater reuse is considered one of the most important options to manage water shortages [6,9,10]. Treated Wastewater (TW) is defined as "water that has received at least secondary treatment and basic disinfection and is reused after flowing out of a domestic wastewater treatment facility" [11]. The characteristics of wastewater vary greatly according to its origin, and it is important to study when treating and reusing it [12–14]. The safety of TW that is

reused for crop irrigation is a relevant issue worldwide [15]. The reuse of TW could be one of the main alternative options to expand water resources [16], especially in dry areas, because it represents another source of renewable water [17]. Furthermore, the 2017 World Water Development Report highlighted the relevance of water reuse [18]. Therefore, the utilization of TW in the countries that suffer from water shortages is encouraged [19], especially because there are huge amounts of wastewater. For instance, China is considered the largest generator of wastewater. In 2012, approximately 68.5 billion tons of this water was released from industrial and municipal sources, which is equivalent to the yearly stream volume of the Yellow River [20]. About 108.16 billion $m^3$ of wastewater (34.33 billion $m^3$ from domestic sources and 73.83 billion $m^3$ from industrial sources) are being generated annually in China [21]. In Egypt, about 5 billion $m^3$ of sewage water were collected every year [22]. Therefore, the TW can add up to 5 billion $m^3$ to Egypt's water resources. Wastewater treatment and use for irrigation represent a valuable resource and an appealing choice, especially in dry regions, because wastewater is considered a further inexhaustible, reliable, and dependable source of water and nutrients [8,23,24]. Therefore, in several water-scarce countries worldwide, wastewater reuse is considered a long-established practice and very important [25]. Potential wastewater reuse applications include agricultural and landscape irrigation, groundwater recharge, industrial reuse, urban applications such as street cleaning, and firefighting and ecological and recreational uses [25,26]. However, the reuse of wastewater for agricultural irrigation is more acceptable than its reuse in other fields [6,27,28]. Recently, TW irrigation has gained a high degree of importance, particularly in dry regions [29–32]. Most countries do not have rules to control wastewater reuse, and, in contrast, many countries have very strict regulations. There are no significant constraints on using the secondary TW as a fertigation source [33]. Besides the decrease in using freshwater, wastewater reuse has decreased the release of wastes into ecosystems and enhanced the soil with nutrients and organic matter (OM) [34]. Refs. [8,35–37] stated that using TW as an irrigation source has economic and environmental benefits since it could reduce or even eliminate the need to supply expensive chemical fertilizers to the soil. Wastewater has OM and nutrients that are useful to the plants [38,39], and thereby has been recognized as an important resource for an agricultural production increase with low cost [40]. However, there are dangers with reusing the wastewater in agriculture; for example, its use led to a rise in the soil salinity, as well as the existence of microbial microorganisms and pollutants [41]. Moreover, this water can carry pathogens that affect human health, besides raising the risk for parasitic, viral, and bacterial diseases in consumers of crops irrigated with this water [42,43]. TW reuse will not only alleviate the water shortage problem for agricultural development but also remedy the pollution and health hazards related to the indiscriminate disposal of untreated sewage water [44,45]. Untreated or insufficiently treated wastewater can cause public health, environmental, and economic problems [46–48]. Therefore, the correct methods must be followed in the treatment and use of wastewater, particularly because wastewater reuse may cause public health hazards if the treatment is not appropriate [28,49]. Refs. [50,51] showed that the different degrees of conventional treatment are:

(1) Preliminary: Remove the large solid materials from the crude wastewater that are conveyed by sewers that could hinder the discharge or cause damage to equipment, such as wood, rags, fecal material, and heavier grit particles.

(2) Primary: Remove the suspended solids (SS) and floating substances.

(3) Secondary: The secondary treatment process aims to diminish the biochemical oxygen demand (BOD), chemical oxygen demand (COD), and SS, and the set of other harm parameters by removal or reduction in residual settleable solids and floating materials from primary treatment. BOD is the amount of dissolved oxygen needed by aerobic biological organisms in water to break down organic material existing in a water sample at a certain temperature over a specific period [52,53]. The COD represents the quality of oxygen required to stabilize the carbonaceous organic matter chemically [54].

(4)　Tertiary and/or advanced: Removal of nutrients and heavy metals (HM), which are not removed by the previous treatment. Additionally, decreasing the microbiological constituents by using some options such as chlorination, ultraviolet rays, and ozonation in disinfection operation.

In general, the negative impact of TW can be reduced significantly by selecting a proper irrigation system, an appropriate cropping pattern with appropriate and effective irrigation management, as well as continuous examination of water, soil, and plant quality, and by taking careful and precautionary actions against pathogens.

The aim of this writeup is to cast light on the importance of using TW in irrigation as an alternative freshwater source. It also reviews the TW irrigation impacts on soil properties, fertility status, plants, and general health. This includes a review of the irrigation systems used with TW.

## 2. Characteristics of Wastewater

The characteristics of wastewater are broadly classified into physical, chemical, and biological properties [55]. They also stated that the liquid portion of the wastewater comprises a complex mixture of minerals and OM in many forms, including large and small particles, floating suspension, and colloidal. Wastewater has some poisonous elements, for example, arsenic, cadmium, chromium, lead, copper, zinc, mercury, etc. [56]. Among the organic substances existing in this water are pesticides, carbohydrates, fats, proteins, synthetic detergents, pharmaceuticals, and complex nitrogenous OM products [57–60]. These poisonous elements have hazardous effects on general health [42,60,61]. However, direct evidence of adverse human health impacts is still being discussed [57,62]. Microplastics, polymer fibers, polyethylene terephthalate, polyethylene, polypropylene, and polystyrene are present in the wastewater [63]. Microplastics can have an inimical effect on the reproductive and vegetative growth of plants [64]. A large number of studies have stated that TW contains pharmaceutically active compounds (PhACs) [65–68]. Conventional wastewater treatment based on activated sludge could not efficiently remove these compounds. As a result, many of these chemicals were later detected in soils watered with TW [69]. Additionally, they mentioned that the ecotoxicological hazard of PhACs in the soil was very low. However, this is widely dependent on the PhACs, as shown by [70]. The soil microorganism can be affected if these compounds accumulated for many years in the soil and can be moved to the crops and then to the food chain, probably risking humans [71]. The chemical and biological constituents and the physical properties of wastewater and their sources are placed in Table 1. Additionally, the important contaminants of interest in wastewater treatment are placed in Table 2.

**Table 1.** Physical, chemical, and biological characteristics of wastewater and their sources [72].

| Characteristic | Sources |
| --- | --- |
| Physical properties | |
| Color | Domestic and industrial wastes, natural decay of organic materials |
| Odor | Decomposing wastewater, industrial wastes |
| Solids | Domestic water supply, domestic and industrial wastes, soil erosion, inflow infiltration |
| Temperature | Domestic and industrial wastes |
| Chemical constituents: | |
| Organic | |
| Carbohydrates | Domestic, commercial, and industrial wastes |
| Fats, oils, and grease | Domestic, commercial, and industrial wastes |

**Table 1.** *Cont.*

| Characteristic | Sources |
| --- | --- |
| Pesticides | Agricultural wastes |
| Phenols | Industrial wastes |
| Proteins | Domestic, commercial, and industrial wastes |
| Priority pollutants | Domestic, commercial, and industrial wastes |
| Surfactants | Domestic, commercial, and industrial wastes |
| Volatile organic compounds | Domestic, commercial, and industrial wastes |
| Other | The natural decay of organic materials |
| Inorganic | |
| Alkalinity | Domestic wastes, domestic water supply, groundwater infiltration |
| Chlorides | Domestic wastes, domestic water supply, groundwater infiltration, water softeners |
| Heavy metals | Industrial wastes |
| Nitrogen | Domestic and agricultural wastes |
| Acidity | Domestic, commercial, and industrial wastes |
| Phosphorus | Domestic, commercial, and industrial wastes natural runoff |
| Sulfur | Domestic water supply, domestic and industrial wastes |
| Toxic compounds | Industrial wastes |
| Gases | |
| Hydrogen sulfide | Decomposition of domestic wastes |
| Methane | Decomposition of domestic wastes |
| Oxygen | Domestic water supply, surface-water infiltration |
| Biological constituents: | |
| Animals | Open watercourses and treatment plants |
| Plants | Open watercourses and treatment plants |
| Bacteria | Domestic wastes, surface water infiltration, treatment plants |
| Archae | Domestic wastes, surface-water infiltration, treatment plants |
| Protista | Domestic wastes, treatment plants |
| Viruses | Domestic wastes |

**Table 2.** Important contaminants of concern in wastewater treatment [72].

| Contaminants | Reason for Importance |
| --- | --- |
| Suspended solids | Suspended solids can lead to the development of sludge deposits and anaerobic conditions when untreated wastewater is discharged into the aquatic environment. |
| Biodegradable organics | Composed principally of proteins, carbohydrates, and fats, biodegradable organics are measured most commonly in terms of BOD (biochemical oxygen demand) and COD (chemical oxygen demand). If discharged untreated to the environment, their biological stabilization can lead to the depletion of natural oxygen resources and the development of septic conditions. |

**Table 2.** *Cont.*

| Contaminants | Reason for Importance |
| --- | --- |
| Pathogens | Infectious diseases can be transmitted by the pathogenic organisms in wastewater. |
| Nutrients | Both nitrogen and phosphorus, along with carbon, are essential nutrients for growth. When discharged to the aquatic environment, these nutrients can lead to the growth of undesirable aquatic life. When discharged in excessive amounts on land, they can also lead to the pollution of groundwater. |
| Refractory organics | These organics tend to resist conventional methods of wastewater treatment. Typical examples include surfactants, phenols, and agricultural pesticides. |
| Heavy metals | Heavy metals are usually added to wastewater from commercial and industrial activities and may have to be removed if the wastewater is to be reused. |
| Dissolved inorganic solids | Inorganic constituents such as calcium, sodium, and sulfate are added to the original domestic water supply as a result of water use and may have to be removed if the wastewater is to be reused. |

The chemical risks associated with wastewater usage are that it contains HM, OM, salt, nutrients, and toxic compounds [73,74]. The chemical composition of wastewater is more varied and more concentrated and contains certain various acids, alkalis chemical contaminants, oil, coarse solids, and other constituents. Inorganic constituents include high concentrations of calcium, sodium, potassium, chlorine, phosphate, sulfur, bicarbonate, ammonium salts, and HM [75,76]. TW contains many of the micronutrients that the plant needs, such as copper, iron, manganese, zinc, boron, molybdenum, cobalt, and nickel [77]. Ref. [78] mentioned that the chemical characteristics of wastewater could adversely affect the environment in many different ways. Soluble organic can deplete oxygen levels in the stream and give taste and odor to water supplies, as well as toxic materials that can affect the food chain and public health.

Ref. [55] reported that the properties of wastewater that can be identified by the human sense organs are termed the physical characteristics. The most important physical characteristics of wastewater are its solid content, as it affects the water's aesthetics, clarity, and color. Other physical parameters are temperature and odors and are not commonly altered in a wastewater treatment plant. The temperature of wastewater is important primarily because it affects aquatic and biological life in the receiving body of water. Higher temperatures decrease the dissolved oxygen solubility in the water. Ref. [79] reported that SS might cause the undesirable conditions of increased turbidity and silt load in the receiving water. A considerable amount of dissolved solids may be added to water during its treatment and use. The term biological characteristics of water refers to the aquatic life and viruses found in water. The quality of water is significantly affected by these characteristics. Algae, for example, cause taste and odor. Some types of algae clog sand filters; others produce nuisance-causing slimy growths on equipment, tanks, and reservoir walls [80]. Furthermore, some microalgae produce powerful toxic substances harmful to living organisms [81]. Microbiological life in water, such as bacteria, viruses, and protozoa, can cause different diseases [82]. The environment of wastewater considers an ideal environment for growing viruses, bacteria, and protozoa. The majority is harmless, but sewage also contains pathogenic microorganisms [83,84]. Several researchers have indicated that biological oxidation systems that occur in the secondary treatment of sewage can remove most pathogenic bacteria from sewage [85,86]. Biological wastewater characteristics can be derived with the help of measuring both COD and BOD [72]. Organic waste usually requires oxygen for rapid and effective biological decomposition. Therefore, the greater the number of organic pollutants, the greater the oxygen demand. Hence, higher

BOD and COD indicate higher pollutant content in wastewater. The BOD/COD ratio is widely used to decide the biodegradability of the wastewater [87,88]. After wastewater treatment, the wastewater concentrations of BOD and COD decrease dramatically due to a notable reduction in biodegradable OM in TW [88]. The BOD value is expressed in milligrams of oxygen consumed per L of a sample during five days of incubation at 20 °C, and it considers a mirror of the level of water organic pollution and an indication for the possibility of polluted water or effluent to oxygen consumption [89]. Ref. [90] stated that the master sources of OM affecting the BOD concentration are raw sewage wastewater and industrial wastes. Additionally, they stated that unpolluted water typically has BOD values of 2 mg $L^{-1}$, and the raw sewage has a BOD value of about 600 mg $L^{-1}$, whereas treated sewage effluents have BOD values ranging from 20 mg $L^{-1}$ to 100 mg $L^{-1}$ according to the treatment level. Industrial wastes may have BOD values up to 25,000 mg $L^{-1}$. The COD test is an indirect indicator of organic compounds' contents in water, and it is commonly used to measure the sensitivity to oxidation of the organic and inorganic compounds that exist in water bodies and effluent from sewage and industrial plants [91,92]. It is expressed in milligrams of oxygen per L of water (mg $L^{-1}$) [93]. COD is a useful, rapidly measured variable for many industrial wastes. The ratio between COD and BOD will vary depending on the characteristics of the wastewater [94]. This ratio has been commonly used as an indicator for biodegradation capacity [55].

## 3. Reuse of Treated Wastewater

TW refers to municipal wastewater that has been treated to meet specific water quality criteria with the intent of being used for beneficial purposes [95]. The worldwide wastewater releases around 0.4 trillion $m^3$ per year and contaminating around 5.5 trillion $m^3$ of water each year [96]. Therefore, all countries should be concerned about treating these large quantities of wastewater, and then reuse it. The increasing demands for domestic water due to population growth, improvement in living standards, and the growing industrial sector will raise the amount of wastewater produced, promoting TW reuse worldwide [97]. The major problems associated with this matter include public health and ecological perils plus technical, institutional, socio-cultural, and sustainability aspects [98]. Thus, wastewater treatment and its usage will be highly essential. So, for example, the treatment rate of wastewater in China (the ratio of the TW amount to the total discharge amount of wastewater) increased to 86% in 2014, 3.4 times that of 1999 [99]. Internationally, the TW irrigation has increased around 10–29% per year in Europe, China, and the US, and approximately 41% in Australia [100]. Table 3 shows the whole wastewater generated, gathered, treated, and utilized for irrigation in some countries [101]. The main treatment target is to supply TW with an appropriate and secure level of risk for the environment and public health, and this happens by reducing SS and OM plus removing wastewater chemical and biological constituents that might be harmful to crops and general health [50,99,102].

Agricultural irrigation represents the largest currently TW user globally; hence, this offers significant future opportunities for water reuse in both industrialized and developing countries [103]. In about 44 countries worldwide, 15 million $m^3$ of TW are reused daily for crop irrigation [104]. Ref. [105] stated that 20 million hectares in nearly 50 countries are irrigated with wastewater. Besides the beneficial effects, reusing TW also may negatively affect the environment and health because the pollutants can remain in the TW after the treatment processes and can easily accumulate in all living materials through the food chain [106–108]. Environmental pollution and increased health risks are considered the main disadvantages of reusing wastewater [109,110]. However, effective irrigation management can reduce the negative effect of TW to a manageable level for the environment and public health [50,111]. Ref. [112] indicated that wastewater reuse may lead to health hazards if the wastewater is not appropriately pre-treated (i.e., inadequate pathogen reduction and heavy metal entry into the food chain). So, all risks must be at acceptable levels. The constituents of concern in wastewater treatment and TW irrigation were given in a table in the paper of Asano and Pettygrove [113]. In general, wastewater could be divided into

the water as the bulk volume and three other categories including: SS, colloidal materials, and dissolved materials. Ref. [50] demonstrated that for effective management, the main elements of concern when irrigating with TW are: SS, since filtration might be required especially with the system of micro-irrigation; nutrients to adjust fertilization; salinity to estimate leaching fraction and select the appropriate cropping pattern; and pathogens for careful and precautionary actions, and selecting a proper irrigation system.

**Table 3.** The wastewater generation, collection, treatment, and reuse for irrigation of crops in some countries in relation to the total cultivated area [101].

| Country | Total Area (1000 ha) | Agricultural Area (1000 ha) | Total Agri-cultural Area (%) | Generated Municipal Wastewater ($10^9$ m$^3$ year$^{-1}$) | Collected Municipal Wastewater ($10^9$ m$^3$ year$^{-1}$) | Treated Municipal Wastewater ($10^9$ m$^3$ year$^{-1}$) | Treated Wastewater Used for Irrigation ($10^9$ m$^3$ year$^{-1}$) |
|---|---|---|---|---|---|---|---|
| Australia | 774,122 | 47,307 | 6.11 | - | - | 2 | 0.28 |
| Brazil | 851,577 | 86,589 | 10.1 | - | - | 3.1 | 0.008 |
| China | 960,001 | 122,524 | 12.7 | 48.51 | 31.14 | 42.37 | 1.26 |
| Germany | 35,738 | 12,074 | 33.7 | - | 5.287 | 5.213 | 5.183 |
| India | 328,726 | 169,360 | 51.5 | - | - | 4.416 | - |
| Italy | 30,134 | 9121 | 30.2 | 3.926 | - | 3.902 | 0.087 |
| Jordan | 8932 | 322 | 3.6 | - | 0.115 | 0.113 | 0.103 |
| Pakistan | 79,610 | 31,252 | 39.2 | 3.06 | - | - | - |
| South Africa | 121,909 | 12,913 | 10.5 | 3.542 | 2.769 | 1.919 | - |
| Turkey | 78,535 | 23,944 | 30.4 | 4.297 | - | 3.483 | - |
| UK | 24,361 | 6279 | 25.7 | 4.089 | 4.048 | 4.048 | - |
| USA | 983,151 | 157,205 | 15.9 | 60.41 | 47.24 | 45.35 | - |
| Canada | 998,467 | 50,846 | 5.09 | 6.613 | 5.819 | 5.632 | - |
| Sweden | 44,742 | 2608 | 5.82 | 0.671 | - | 0.436 | - |

## 4. Guidelines and Quality Criteria for Treated Wastewater Reuse in Agriculture

With the progress of treatment technologies, several guidelines have been issued in various countries to provide quality standards and guidance using TW in irrigation and different fields in order to guarantee safe reuse for all environmental aspects [42,114–123]. Ref. [124] presented the guidelines and criteria of water reuse in various countries. In a study conducted to collect and assess the standards and guidelines of reusing water in agriculture worldwide, [125] investigated 70 guidelines and regulations in different countries around the world. The outcomes indicated that the guidelines and regulations are primarily human-health focused, deficient in regards to some of the possible hazardous pollutants, and with huge inconsistencies in the data. Additionally, [124] suggested that the primary reason for existing guidelines for using treated wastewater is to manage adverse effects on people and the environment. In order to manage the public health hazards that arise from the use of wastewater for irrigation, and facilitate the rational use of this water, the World Health Organization (WHO) has given three guidelines in 1973, 1989, and 2006 for the safe use of TW in irrigation [42,126–128]. The main purpose of these guidelines is to advance the standards and support the government regulations relating to wastewater management [42,129,130]. The FAO has also released two guidelines relevant to the use of TW in irrigation. The first classified irrigation water into three groups based on sodicity, salinity, toxicity, and different hazards, as shown in Table 4. [131]. This classification reveals the potential crop-production problems linked with the use of conventional water sources.

In the second guideline, the FAO divided the application of water reuse in irrigation into three groups based on irrigated crop type [56]. The Environmental Protection Agency (EPA) released four water reuse guidelines in 1980, 1992, 2004, and 2012. The last version is considered an update of the 2004 guideline and aimed to advance wastewater reuse by serving as reliable references for reusing water depending on a compilation of global experiences [114,123]. Generally, the new guideline is stricter than its previous guideline with regard to preserving the environment and health [132]. The WHO, EPA, and FAO guidelines are considered the bases for creating the regulations in different countries worldwide. So, if no national guidelines are available in any country, WHO, EPA, and FAO guidelines are suggested as a solution. An example of national guidelines issued by some countries is the Chinese water quality standard for farmland irrigation reuse (GB20922-2007). This standard was enacted by the Ministry of Environmental Protection, and the General Administration of Quality Supervision, Inspection, and Quarantine (Table 5) [117]. Later, a standard for using TW in irrigated landscape (GB/T 25499—2010) was released in 2010 [115], and it suggested stricter limitations for BOD and FC because of the high-exposure risk of TW during irrigation (Table 5) [115].

**Table 4.** FAO guideline for the quality of water used for irrigation [131].

| Possible Irrigation Problem | | Units | Degree of Restriction on Use | | |
|---|---|---|---|---|---|
| | | | None | Slight to Moderate | Severe |
| **Salinity (Affects Crop Water Availability)** | | | | | |
| **ECw** [1] | | dS m$^{-1}$ | <0.7 | 0.7–3 | >3 |
| (or) | | | | | |
| Total dissolved solids (TDS) | | mg L$^{-1}$ | <450 | 450–2000 | >2000 |
| **Infiltration** (affects infiltration rate of water into the soil. Evaluate using EC$_w$ and SAR together) | | | | | |
| SAR | =0–3 and EC | | >0.7 | 0.7–0.2 | <0.2 |
| | =3–6 | | >1.2 | 1.2–0.3 | <0.3 |
| | =6–12 | | >1.9 | 1.9–0.5 | <0.5 |
| | =12–20 | | >2.9 | 2.9–1.3 | <1.3 |
| | =20–40 | | >5 | 5–2.9 | <2.9 |
| **Specific Ion Toxicity** (affects sensitive crops) | | | | | |
| Sodium (Na) | | | | | |
| surface irrigation | | SAR | <3 | 3–9 | >9 |
| sprinkler irrigation | | me L$^{-1}$ | <3 | >3 | |
| Chloride (Cl) | | | | | |
| surface irrigation | | me L$^{-1}$ | <4 | 4–10 | >10 |
| sprinkler irrigation | | me L$^{-1}$ | <3 | >3 | |
| Boron (B) | | mg L$^{-1}$ | <0.7 | 0.7–3 | >3 |
| Trace Elements | | | | | |
| **Miscellaneous Effects** (affects susceptible crops) | | | | | |
| Nitrogen (NO$_3$-N) [2] | | mg L$^{-1}$ | <5 | 5–30 | >30 |
| Bicarbonate (HCO$_3$) (overhead sprinkling only) | | me L$^{-1}$ | <1.5 | 1.5–8.5 | >8.5 |
| pH | | | Normal Range 6.5–8.4 | | |

[1] EC$_w$ is water electrical conductivity in deci Siemens per meter (dS m$^{-1}$) at 25 °C. [2] NO$_3$-N means nitrate-nitrogen reported in terms of elemental nitrogen (NH4-N and Organic-N should be included when wastewater is being tested).

**Table 5.** Chinese water quality standard for TW irrigation [115,117].

| Guideline | Units | Chinese standard for Agricultural Irrigation (2007) | Chinese Standard for Landscape Irrigation (2010) |
|---|---|---|---|
| Sodium adsorption ratio (SAR) | - | - | ≤9 |
| Total dissolved solids (TDS) | mg L$^{-1}$ | ≤1000; ≤2000 [a] | ≤1000 |
| Suspended solids (SS) | mg L$^{-1}$ | ≤60; ≤80; ≤90 [b] | |
| pH | - | 5.5–8.5 | 6–9 |
| BOD | mg L$^{-1}$ | ≤40; ≤60; 80; ≤ 100 [b] | ≤20 |
| COD | mg L$^{-1}$ | ≤100; ≤150; ≤180; ≤200 [b] | |
| Intestinal nematodes | Eggs L$^{-1}$ | ≤2 | ≤1; ≤2 [e] |
| Fecal coliforms (FC) | CFU 100 mL$^{-1}$ | ≤2000; ≤4000 [c] | ≤20; ≤100 [e] |
| Heavy metals (HM) and specific ion toxicity | | | |
| Chloride (Cl) | mg L$^{-1}$ | ≤350 | ≤250 |
| Sulfide (S) | mg L$^{-1}$ | ≤1 | |
| Chlorine residual | mg L$^{-1}$ | ≤1; 1.5 [d] | 0.2–0.5 |
| Petroleum | mg L$^{-1}$ | ≤1; ≤10 | |
| Hydrargyrum (Hg) | mg L$^{-1}$ | ≤0.001 | ≤0.001 |
| Cadmium (Cd) | mg L$^{-1}$ | ≤0.01 | ≤0.01 |
| Arsenic (As) | mg L$^{-1}$ | ≤0.05; ≤0.1 [d] | ≤0.05 |
| Chromium (Cr) | mg L$^{-1}$ | ≤0.1 | ≤0.1 |
| Lead (Pb) | mg L$^{-1}$ | ≤0.2 | ≤0.2 |

[a] Irrigation under saline-alkali soils and non-saline alkali soils. [b] Values for vegetables, cereals cultivated in paddy fields, cereals cultivated in dry farmland, and fiber crops, respectively. [c] Values for vegetables and other crops, respectively. [d] The former value corresponds to vegetables and cereals cultivated in paddy fields, while the latter value corresponds to cereals cultivated in dry farmland and fiber crops. [e] Values for the non-restricted greenbelt and the restricted greenbelt, respectively.

Irrigation water quality standards include several specific characteristics of water that are relevant to maintaining soil fertility, crop productivity, crop quality, and environmental protection [133]. Additionally, they mentioned that parameters such as pH, EC, BOD, COD, Total Suspended Solids (TSS), the Sodium Adsorption Ratio (SAR), adjusted SAR (adj SAR), the Soluble Sodium Percentage (SSP), the Exchangeable Sodium Percentage (ESP), Residual Sodium Carbonate (RSC), and the content of the toxic elements are utilized to evaluate the propriety of water for irrigation purpose. Additional irrigation water quality parameters, which can affect soil characteristics or plant growth, include biological indicators (fecal or total coliforms (FC or TC), *Escherichia coli* (*E. coli*), and helminth eggs), nutrient levels, boron concentration, heavy metal content, and phytotoxic compounds content [134]. Therefore, continuous monitoring of these parameters is recommended, as that may affect the soil, plants, and the environment. According to [50], the main water parameters that are utilized to determine and manage the quality for irrigation purposes could be summarized into the following:

*4.1. Salinity*

The amount and types of salts existing in TW are important to evaluate the appropriateness of TW for irrigation. The level of salinity in TW is always higher than it is in the source [8,74] and depends on the wastewater source and treatment type [135]. The salinity level in TW is normally 1.5–2 times higher than the salinity level of freshwater [136]. This abundance of salts in TW could prompt an increase in the soil salinization, sodication, and structural changes and might lead to a yield decrease. Ref. [7] concluded that in dry areas, salinity and sodicity are considered the main ecological threats that may happen when using TW for irrigation. Ref. [137] reported that ECw is one of the quality parameters necessary to be consistently measured by the farmers to assess the soluble salts amounts in TW. Accordingly, the crop pattern and field management plan will be determined. To minimize the salinity problem, it is necessary to consider the following: selection of salt-tolerant crops; selection of the irrigation system; schedule of irrigation; leaching; installation of

drainage facilities; adequate groundwater depth and land leveling; use of the wastewater in conjunction with freshwater [138–146].

### 4.2. Alkalinity

Wastewater contains soluble salt, which could raise the sodium ions or increase soil salinization [34]. TW is higher in $Na^+$, $Ca^{2+}$, and $Mg^{2+}$, and an increase in the SAR can cause increases in soil alkalinity, and consequently cause severe permeability problems in the soils that are irrigated with TW [147]. Hence, it is necessary to observe the value of the SAR because it is considered the best index to estimate physicochemical variations in the soil [137,148]. In the park and farmland soils irrigated with TW, [149] the soil EC values increased by 12.4% and 84.2%, while the SAR increased by 64.5% and 145.8% compared to the control treatments (drinking water or groundwater irrigation), respectively. In this situation, it is advised to use chemical amendments such as gypsum and to use organic manure and straw as an OM. Additionally, [150] recommended using the drippers or mini sprinkler with a low discharge rate and sufficient time, because this reduces the soil crust formation (that occurs from irrigation with water with a high content of SAR) which allows water to seep into the soil.

### 4.3. Crop Nutrients in Treated Wastewater

Wastewater and TW contain nutrients necessary for plant growth such as N, P, and K, and micronutrients such as Fe, Mn, Zn, and Cu, and a significant amount of OM [151,152]. This makes the TW rich in fertilizer that can enhance soil fertility and increase the production of crops [37,73,153–155]. However, poor management of TW irrigation could result in the accumulation of nutrients in the plants beyond their required level, and this will contribute to environmental and health problems [7]. The content of N, K, and P of municipal wastewater following secondary treatment ranges from 20 to 60 mg $L^{-1}$, 10 to 30 mg $L^{-1}$, and 6 to 15 mg $L^{-1}$, respectively. So, evaluations of NPK in the TW should be made in conjunction with soil testing for fertilization planning [50,134]. Nutrient concentration in TW can be too much for crops, therefore leading to over-fertilization, nutrient leaching, and a decrease in crop size [156]. Supplying plants with nitrogen exceeding their exact requirements through the use of wastewater could cause a reduction in economic yield and food nutrient quality [73,136], and may affect soil microbial communities [8]. Ref. [157] stated that the level of TW nitrogen after secondary treatment in Taiwan is high (15–20 mg $L^{-1}$), which makes it inappropriate for rice cultivation. When using TW irrigation, the improvement in crop production is associated with crop type and nutrients' levels in the TW. Therefore, it is essential to estimate the quantities of nutrients in TW and soil because these quantities must be deducted from the amount of fertilizer required by the plant.

### 4.4. Heavy Metals and Specific Ion Toxicity

Heavy metals that exist in municipal wastewater (such as Arsenic (As), copper (Cu), Cadmium (Cd), nickel (Ni), molybdenum (Mo), Chromium (Cr), zinc (Zn), and Lead (Pb)) are effectively removed by the correct treatment processes until their concentrations are near to the concentrations in freshwater or are within the permissible limits [102,158–161]. However, [106] stated that the HM could remain in TW and accumulate in soils, and that might be risky, especially if the source is industrial wastewater or mixed between industrial and domestic wastewater [140,162,163]. Entry of these metals into the food chain should be taken into consideration when using this water for a long time [164]. Besides reducing the HM content in wastewater by the treatment process, diluting the TW with fresh water can also reduce the level of these metals [165]. However, significant HM concentration has been noticed in the top layers of soil that were irrigated with TW for 20 years [166]. Ref. [136] illustrate that in Beijing, the concentration level of some HM in TW was similar to the concentration in the groundwater (GW), except for Zn, as shown in Table 6.

**Table 6.** The concentration of some HM in GW and TW, and cropland soils irrigated with GW and TW, respectively [a] [136].

| | Concentration in Water (µg L$^{-1}$) | | Concentration in Cropland Soils (mg L$^{-1}$) | |
| --- | --- | --- | --- | --- |
| | GW [a] | TW [a] | GW Irrigated | TW Irrigated |
| As | 1.92 | 1.5 | 9.51 | 8.09 |
| Cd | 0.022 | 0.024 | 0.13 | 0.16 |
| Cu | 2.66 | 4.47 | 29.17 | 20.55 |
| Cr | 1.84 | 1.13 | 56.17 | 57.2 |
| Pb | 0.75 | 1.02 | 15.9 | 17.7 |
| Zn | 12.9 | 29.2 | 65.31 | 53.03 |

[a] GW and TW stand for groundwater and treated wastewater, respectively.

The high concentration of HM in the wastewater can cause hazards for humans and animals due to the accumulation of these metals in soils and crops [167]. However, HM that exist in wastewater used in irrigation may not be a serious problem if these metals are within a safe limit, but when their quantity exceeds the limit, they become toxic [168]. Therefore, when using TW irrigation, it is necessary to monitor HM concentrations in water, soils, and crops because high levels of these compounds could accumulate these compounds in the crops, which is considered a risk to human and animal health. Ref. [56] gave a recommendation for the maximum concentrations for some trace elements in irrigation water such as Mn, Fe, Zn, Pb, Cu, Cd, and Ni of about 0.2, 5.0, 2.0, 5.0, 0.2, 0.01, and 0.2 mg L$^{-1}$, respectively. Recommended limits for constituents in TW use for irrigation are presented in Table 7 [169].

**Table 7.** Recommended limits for constituents in TW for irrigation [169].

| Constituent | Long-Term Use [a] (mg L$^{-1}$) | Short-Term Use [b] (mg L$^{-1}$) |
| --- | --- | --- |
| Aluminum (Al) | 5 | 20 |
| Arsenic (As) | 0.10 | 2 |
| Beryllium (Be) | 0.10 | 0.50 |
| Boron (B) | 0.75 | 2 |
| Cadmium (Cd) | 0.01 | 0.05 |
| Chromium (Cr) | 0.10 | 1 |
| Cobalt (Co) | 0.05 | 5 |
| Copper (Cu) | 0.20 | 5 |
| Fluoride (F$^-$) | 1 | 15 |
| Iron (Fe) | 5 | 20 |
| Lead (Pb) | 5 | 10 |
| Lithium (Li) | 2.50 | 2.50 |
| Manganese (Mg) | 0.20 | 10 |
| Molybdenum (Mo) | 0.01 | 0.05 |
| Nickel (Ni) | 0.20 | 2.00 |
| Selenium (Se) | 0.02 | 0.02 |
| Vanadium (V) | 0.10 | 1 |
| Zinc (Zn) | 2 | 10 |

[a] For water used continuously on all soils. [b] For using the water for a period less than 20 years on alkaline or fine-textured neutral soils.

With appropriate and effective irrigation management, the negative impact of TW can be reduced significantly, allowing it to be used widely [170]. In general, continuous examination of water, soil, and plant quality is required to ensure the successful use of TW [171].

## 5. Soil Properties as Affected by Irrigation Water Quality

### 5.1. Physical Properties

Wastewater irrigation alters the microbiological and physicochemical properties of the soil [8]. Ref. [172] announced that depending on the amount of OM in the wastewater used for irrigation, soil OM will increase, and that increases the ability of soil to hold water, consequently affecting the compaction resistance and drainage properties. There is no consensus in the scientific literature on the effects of irrigation with wastewater on soil physical properties; for example, many studies agree that the soils irrigated with wastewater have a notable decline in hydraulic conductivity [173–177]. In contrast, other studies showed an improvement in soil hydraulic conductivity [178,179]. Ref. [180] pointed out that urban wastewater irrigation leads to increased soil porosity and hydraulic conductivity. Additionally, [181] stated that domestic wastewater does not negatively affect hydraulic parameters. Ref. [182] found that when the soil was irrigated with wastewater, the soil bulk density increased, and that was explained by the accumulation of OM. Contrary to this, [172] found a slight decline in the bulk density of soil irrigated with wastewater for a long time. Ref. [183] studied the effect of two irrigation water qualities on some physical properties of fine sandy loom soil (The ECw and SAR values were 1.5 dS m$^{-1}$ and 4.5 in the first type vs. 12 dS m$^{-1}$ and 11.0 in the second one). They found that the values of water retention and bulk density of the soil were not considerably affected by the two studied water qualities. Ref. [184] mentioned that the bulk density values of clay, loamy, and calcareous soils were decreased, while hydraulic conductivities were relatively increased when the irrigation water salinity increased from 1500 to 4000 mg L$^{-1}$. In a study carried out by [185], there was a decline in infiltration with time for the treatments that received freshwater and TW. Ref. [186] examined the wastewater irrigation effect on soil moisture characteristic curves and found, as a general trend, that the percentages of all soil moisture contents increased due to sewage water application. These results might be ascribed to the rise in both fine fractions and OM contents. Ref. [187] indicated that TW irrigation increased soil electrical conductivity and soil water content. Refs. [188,189] found that the soil water holding capacity was significantly increased when the soil was irrigated with wastewater. They also reported that using this water for a long term led to a gradual rise in the maximum water-holding capacity, and improved the physical conditions of most soils.

### 5.2. Chemical Properties

Many studies have been conducted to examine the impacts of wastewater irrigation on soil chemical characteristics such as salinity, sodicity, pH, etc. Salinity is evaluated via EC and SAR, which together refer to the level of soil saturation with sodium and problems with infiltration [190]. Ref. [191] conducted a study to examine the effects of using secondary TW in irrigation. The results showed that when applying TW, there were no significant effects on soil physicochemical characteristics compared with freshwater, except for EC and SAR, which were slightly higher in TW soil samples. Ref. [192] stated that when irrigated with TW, the soil sodium levels increased by the end of the irrigation period but declined throughout the wet season. Additionally, [193] reported that irrigation with TW caused increased soil sodification. Ref. [194] noted that the SAR value was high in the soils of farmland that received TW for about one decade. Ref. [195] also found that wastewater irrigation led to increasing soil salinity. Ref. [196] found that increasing irrigation water salinity from 0.58 to 3.67 dS m$^{-1}$ increased soil salinity from 1.80 to 24.83 dS m$^{-1}$. Thus, the cause of increasing soil salinity was the salt concentration of irrigation water. Ref. [197] assessed the impact of TW on the soil characteristics and noticed that SAR, salinity, and the organic content of soil increased as a result of high evaporation rates, low rainfall, and the absence of drainage systems. Ref. [197] assessed the impact of TW on the soil characteristics and noticed that SAR, salinity, and the organic amount of soil were higher as a result of high evaporation rates, low rainfall, and the absence of drainage systems. However, using TW irrigation had no significant impact on the salinization or sodicilization of soil [198]. Additionally, [199] indicated that after infiltration and evaporation of TW, the value of

soil SAR demonstrated a low possibility of soil alkalization. Similar results were also supported by other studies conducted in China [99]. Therefore, the inconsistent results in the impacts of TW irrigation on salinization and alkalization of soils could be a result of the TW irrigation period and the salts' concentration in this water [200]. Ref. [201] stated that soil salinity under TW treatment was higher when compared with freshwater treatment. However, there was no increase in leaf Cl or Na concentrations, nor a decrease in tree productivity. Nevertheless, they noted that prolonged irrigation with TW could adversely impact soil physicochemical characteristics. Ref. [202] revealed that the TW increased soil EC and decreased the soils' capacity for holding nutrients. Ref. [203] reported that the capacity of the cation exchange was increased by the increasing wastewater applied due to the increasing finer materials and organic content. The high BOD in sewage effluents or treated sewage increased the soil OM content [204]. Refs. [183,205] found that soluble sodium in soil was significantly increased due to increasing salt concentrations in irrigation water. Irrigation with TW for the long term can raise salinization of the soil. Therefore, the soil sodium content must be observed using TW [206]. Ref. [207] studied the TW irrigation impact on soil characteristics, and they reported that the irrigation of clayey and sandy soils with TW for the long term led to a rise in soil salinity, but it declined with each rainfall. Ref. [208] indicated that the main effect of TW irrigation was the increase in electrical conductivity in the lower soil horizons. Additionally, they observed that Al, Fe, and Zn showed a twofold to eightfold accumulation in the topsoil layer after two years. Ref. [192] showed that when the soil received TW, the concentrations of HM and hydrocarbons in the soil are similar for all irrigation treatments. However, [209] reported that the HM in TW tend to accumulate in soil and become bioavailable. Ref. [8] stated that the soil pH raised after 60 years of irrigation with secondary-treated municipal wastewater. In contrast, some studies indicated that irrigation with wastewater caused a notable decline in soil pH with prolonged wastewater irrigation [210,211]. Ref. [212] reported that the TW irrigation increased the pH level in the top layer of soil compared to the control treatment. Additionally, [189] stated that soil pH decreased after applying wastewater irrigation, and $CaCO_3$ content was slightly influenced. The author added that applying this water for up to eight years caused a decline in soil salinity due to the washing out of most salts from the virgin sandy soils; after that, there was a gradual rise in soil salinity with the length of irrigation period with wastewater. Ref. [213] measured the chemical characteristics of silty clay soils irrigated with low-quality water, and the study showed that the $CaCO_3$ contents, OM contents, pH value, EC value, cation exchange capacity (CEC), and exchangeable sodium percentage (ESP) were relatively higher than in the soils irrigated with non-polluted water. She concluded that the high ESP values (mean, 20.59), besides the relatively high $CaCO_3$ content (mean, 4.0 %), increased the pH values of these soils. The most important reason for increasing $CaCO_3$ content in the polluted soils (mean, 3.1–4.0%) was the high $HCO_3^-$ ion concentration in the polluted irrigation water. Ref. [214] examined the impacts of the wastewater irrigation period on the changes of some chemical characteristics under two different plants. They noticed that the pH values and the content of $CaCO_3$ for both rhizosphere and bulk soil decreased as the irrigation period increased. Additionally, they noticed that the contents of HM in both rhizosphere and bulk soils of the two plants increased as the irrigation period increased. Recently, using TW or low-quality water became a part of the extension program for maximizing water resource usage. However, the uncontrolled application of such waters has many unfavorable effects on soils and plants grown, especially in long-term use. The hazard effects are mainly related to the water quality and soil characteristics, besides the types of growing crops [215].

*5.3. Biological Properties*

5.3.1. Soil Enzymes

Microbiological properties, including enzyme activities, are potentially helpful as an index of soil quality [216], as these properties respond speedily to soil management, fertilization processes, and the HM concentration [217]. The soil microbial community is

important in regulating the material cycle of the ecosystem [218]. Soil microorganisms play an essential role in decomposing organic matter, cycling nutrients, fertilizing the soil, and developing the soil structure, since the soil microorganisms produce soil organic carbon that improves soil fertility and water-retaining capacity [219]. Soil enzymes are responsible for catalyzing soil biochemical reactions necessary for microbial life functions. Enzymes increase the reaction rate at which organic matter decomposes and releases nutrients into the soil environment [220]. Soil enzymes are responsible for the biogeochemical cycling of many elements, and their activities reflect the extent of chemical and biological reactions in soils [221,222]. Therefore, "inhibition of the activities of one or more key enzymes could have significant consequences in the rate of elemental cycling and, consequently, the long-term sustainability of receiving soils could be jeopardized" [221]. The enzymes play a crucial role in transforming nutrients and increasing the accessibility of nutrients in the soil [218]. Soil enzymes perform essential biochemical functions in material and energy conversion in the soil ecosystem, including mineralization, OM degradation, and nutrient recycling [99]. In addition, it helps to describe and make predictions on the interaction between ecosystems. The most valuable use of soil enzymes is to evaluate the impacts of various activities and chemicals on the soil's relative health [223]. Soil enzyme activities are sensors of soil microbial status and soil physicochemical conditions [224,225]. Ref. [226] mentioned that the soil enzymes are considered an indicator of the effects of the soil treatments on the fertility of soils. Therefore, the soil microbial component and soil enzyme activities are indices for monitoring various impacts on soils due to their role in the soil environment. As mentioned before, TW contains nutrients, salts, HM, and organic pollutants. All these could change the enzyme activity in soils. Ref. [227] stated that there are connections between the activity of soil enzymes and soil organic carbon, phosphorus, and nitrogen. Ref. [209] reported that the soil enzyme activities could be improved when the soil received TW. Additionally, [228] mentioned that TW irrigation helps increase the soil enzyme activity, which can lead to improved plant growth and yields. Ref. [229] observed a positive impact on alkaline phosphatase and β-glucosidase activity, which refers to improvement in soil-biological and physicochemical properties. TW contains nutrients and OM, which can enhance the soil's biological health [136]. Refs. [230,231] stated that the activity of enzymes has a positive correlation with the soil nutrients, but this correlation is negative with salts. Additionally, the correlation is negative with the rise in HM in the soil [107,232,233]. Nevertheless, the positive effect associated with nutrients in the soil is relatively greater than the negative effect that is linked with the increase in HM [194]. Ref. [136] reported that the irrigation with TW for the long term in two sites in Beijing and California led to improved soil-biological activities. In an experiment conducted in Beijing, [194] found that compared with soils which received freshwater, the irrigation with TW led to increased soil microbial biomass and improved soil enzyme activities (invertase, alkaline, phosphatase dehydrogenase, urease, and catalase). They found that the enzyme activity in the surface layer (0–20 cm) of soil improved by an average of 36.7% when using TW compared with the freshwater. In five sites irrigated with TW for the long term in California, [221] pointed out that activities of 17 soil enzymes were improved by an average of 2.2-fold to 3.1-fold when compared with the control. Additionally, [223] stated that the soil enzyme activities had shown an important change among soils compared to the reference site, which was never irrigated with TW.

### 5.3.2. Microbial Biomass

Soil microbial biomass is one of the most important soil-biological attributes. This parameter regulates many critical ecosystem processes, including biophysical integration of OM with soil-solid, aqueous, and gaseous phases [234,235]. Therefore, changes in microbial activities and microbial biomass could severely affect some soil processes and could also provide a warning signal of deleterious changes in soil health [236]. Microbial communities are essential for the ecosystem concerning direct interactions with plants and nutrients and OM cycling [237]. Ref. [238] concluded that HM have strong inhibitory

impacts on soil enzymatic activities and the microbial community structure. Ref. [239] found a positive relationship between clay content and the soil microbial biomass content, and there was no significant correlation between soil moisture and microbial biomass [240]. Ref. [187] reported that TW increased the Proteobacteria population in the soil, while clean water tended to increase Acidobacteria numbers in the soil. Additionally, they concluded that ammonia nitrogen (NH4+-N), total phosphorus, and electrical conductivity were the important factors that had a significant impact on the structure of the soil microbial community under TW irrigation. Both enzyme activities and microbial biomass were affected by the crop type [241]. When using TW irrigation, [223] found a significant decline in microbial biomass in short-term-irrigated soils, while a significant increase was recorded in soil irrigated for 20 years with TW. Additionally, [242,243] had proved that microbial activity rose significantly in soils irrigated for a long term with TW. However, [228] reported that a 40-year research project on the impact of TW irrigation on the microorganisms of soil demonstrated that there was no significant difference between soil microbial composition and quantity when the soil was irrigated with TW and groundwater. Additionally, [243] reported that Shannon diversity index values suggested that the microbial diversity was not significantly different between soils irrigated with TW and freshwater. However, they observed that most of the sequences associated with nitrogen-fixing bacteria, carbon degraders, nitrifying bacteria, denitrifying bacteria, potential pathogens, and fecal indicator bacteria were more abundant in TW than in freshwater. Therefore, TW may contain bacteria that may be very active in many soil functions as well as some potential pathogens. Ref. [244] pointed out that different microbial strains including worms, protozoa, bacteria, and viruses could result in serious health risks, especially among those directly exposed to TW or the consumption of vegetables irrigated with wastewater. However, boiling the water before sanitization and rinsing the vegetables help in the reduction and elimination of microbial risks [245,246].

*5.4. Fertility Status*

5.4.1. Macronutrients

The use of TW for irrigation and plant nutrition improves chemical, physical, and soil fertility [8]. TW irrigation could provide soils with OM and nutrients [16], thus improving crop production [212]. Irrigation with TW can enhance soil health when using proper management practices [136,247]. Compared to the soils that had received freshwater irrigation, many researchers illustrated that the effective P, total N, and total K content increased significantly in soils that received TW irrigation [87,187,248,249]. Ref. [250] reported that total N increased in the top layer of soil when irrigated with effluent for eight and 20 years. A similar tendency was observed with available P and available K [172]. However, in an experiment conducted by [251], they observed that there was no obvious rise in soil nitrogen, phosphorus, or potassium after short-term irrigation with TW. These contradictory results may be due to nutrient concentration in the TW or how long this type of water was used for irrigation [99]. Several investigators had studied the correlation between the irrigation water salinity or soil solution and the status of soil macronutrients (N, P, and K). In this regard, [252] reported that the salinity of soil solution has a negative effect on microorganisms' activity that is responsible for organic materials' decomposition and release of available inorganic N. Ref. [253] indicated that the soil irrigated with sewage effluent was higher in available P and K content than in the soil irrigated with the Nile fresh-water. Concerning the effect of water quality on the status of K in soil, [35] mentioned that increasing the irrigation water salinity generally enhanced the K concentration in the soils. Additionally, they mentioned that the irrigation of sandy soils with sewage water for five years led to an increase in the total P content from about 178 to 2820 mg kg$^{-1}$, and the total-N increase with the length of utilization of sewage water.

### 5.4.2. Micronutrients and Heavy Metals

Several studies have been carried out to investigate the impact of TW irrigation on potentially toxic elements' (PTE) contents in soils and crops [254,255]. It was observed that more PTE were concentrated in the topsoil layer irrigated with TW than concentrations with freshwater irrigation [256,257]. Ref. [212] observed that the TW has a slight impact on the PTE contents in soil when the soil had been irrigated with TW for 35 years. Additionally, they mentioned that long-term TW irrigation would cause a significant rise in OM content and might improve the soil quality. The high-organic matter contents in the soil can decrease PTE movement and the effect of leaching [258]. Refs. [259,260] stated that the TW irrigation resulted in increasing the HM (Zn, Pb, Co, Cu, Cd, and Ni) in the soil. This rise in available HM is a considerable obstacle, as it adversely influences characteristics of soil that possibly influence the soil quality. In contrast, [261] did not found a significant variation in the concentrations of HM in soils irrigated with TW and with freshwater in different years. Additionally, [262,263] illustrated that there is no HM accumulation in soils that received TW irrigation in China. Ref. [136] concluded that, in the lands irrigated with TW, the HM do not cause a problem in soil or the food chain. In a field irrigated with TW for 20 years, [250] revealed that the concentrations of Cr, Zn, Cu, and Ni in this field were only slightly higher than those in the field irrigated with groundwater. Ref. [264] noticed that the contents of Pb, Ni, Cd, Cr, and Co within the soil layer of 60 cm increased by extending the sewage irrigation period. These elements' accumulation is more obvious in the upper layers than in the sub-surface. Ref. [265] indicated that soil availability of Zn, Cu, Fe, and Mn increased as a result of sewage irrigation during the three years of study; Co, Cr, Pb, Cd, and Ni slightly increased. Ref. [266] informed that extending irrigation of the soils with the wastewater markedly increased the content of soil OM, the micronutrients (Fe, Zn, Cu, and Mn), and HM (Co, Pb, Cd, and Ni). Ref. [267] declared that sewage water might have a low amount of HM, but using this type of water for a long time could accumulate significant amounts of HM in soil. Ref. [213] investigated the impacts of long-term irrigation with low-quality water on some clayey soils' HM contents. She stated that these soils attained relatively higher contents of Cu, Zn, Mo, Ni, Pb, Co, Cd, and Cr than those irrigated with fresh water. Ref. [268] illustrated that the different amounts of TW have no considerable effect on the HM contents in the soil. However, [269] indicated that the prolonged irrigation with wastewater contaminated with HM increased considerably HM contents of the tested soil. From the previous references, it is concluded that the HM concentration in the soils depends on the irrigation period, whether it is prolonged or shortened, and depends on HM concentration in irrigation water.

## 6. Effect of Wastewater Irrigation on Plant

TW is considered a steady water source that can supply a large amount of nutrients [187,270], and reusing it in irrigating the crops can considerably increase crop production [212,250]. However, wastewater usage for crop irrigation needs careful control due to the potential presence of unwanted constituents in the sludge, such as HM and contaminants [271,272]. In a three-year study, results showed that the nectarines' quality parameters, antioxidant compounds, and total phenolic were higher in the fruits under TW treatment than in the fruits under freshwater treatment due to the huge quantity of nutrients in the TW. However, the fruit number was lower under wastewater treatment, but this reduction was compensated with a large weight for individual fruits [273]. In China, the cereal crop irrigated with TW has increased rapidly since 2002 to solve the water shortage problem [99]. Ref. [274] demonstrated that seven crops (wheat, celery, maize, millet, apples, yellow beans, and rapeseed) were tested. The results illustrated that the production of the crops irrigated with TW was significantly higher than among the crops without irrigation. The rise in the yield of wheat, barley, oat, and mustard due to TW application has also been reported by [275–277]. Additionally, the growth of sugarcane was better under TW irrigation treatment than under the control treatment [278]. In Beijing, using pot experiments, [279] assessed the impacts of using TW and freshwater on

soybean and maize growth. The outcomes revealed that the yield of soybeans and maize under TW treatment was clearly improved. However, [156,280] reported that no noticeable difference between yields of maize and wheat irrigated with conventional water and TW. In another study, [281] stated that under TW irrigation, the yield production of corn was increased, and they attributed this rise to the soil's physical characteristics' improvement and enhanced nutrient uptake. Ref. [282] conducted a study to examine the effect of fresh water and TW irrigation under surface drip irrigation (SDI) and subsurface drip irrigation (SSDI) on growth parameters and the production of okra. The results revealed that TW positively influenced the yield attributes and growth parameters of okra. Additionally, the results revealed that the maximum agronomic performance of okra was recorded with TW by comparing it with freshwater. Ref. [283] informed that wastewater irrigation for two years led to barley biomass increasing. Refs. [277,284] reported that TW irrigation is beneficial because of the huge quantity of nutrients necessary to sustain soil fertility and raise plant growth and productivity. That result might be due to the presence of $NH_4^+$ and $NO_3^-$, the two ionic forms of nitrogen that are effective in raising the meristematic cells' number. Ref. [285] detected that the TW caused an increase in HM in most plant samples, particularly for Cu, Mn, and Zn. However, some plants such as alfalfa and corn that received freshwater had slightly higher values of Cu, Fe, and Mn. The results also indicated that only the Fe amount was higher than the critical limit for both irrigation types. In China and other countries, many researchers illustrated the positive benefits of TW irrigation on the growth and yield of vegetables [249,250,286]. A significant increase in fruity vegetable yield was observed when irrigated with TW. Ref. [287] stated that the yields of tomato, kidney bean, cucumber, and eggplant irrigated with TW were 15, 7, 24, and 61% higher than those irrigated with freshwater, respectively. Ref. [228] indicated that there was a positive influence in the yield and growth of tomato, eggplant, and cucumber irrigated with TW compared with groundwater irrigation, suggesting that TW irrigation can probably replace groundwater irrigation. Ref. [191] mentioned that TW can be a safe, alternative source for leafy and root crops irrigation. Ref. [288] confirm that the maize can save water without affecting crop productivity when irrigated with TW. Ref. [289] showed that sorghum irrigated with TW produced more dry biomass, energy yield, and ethanol than sorghum biomass with freshwater. Tomatoes under irrigation with TW yielded more fruit and achieved higher yields than plants irrigated with tap water [290]. Several researches revealed that the irrigation with TW had no adverse effect on vegetables, groundwater, or the food chain [287,291–294]. Ref. [295] reported that using TW in pepper cultivation did not have a harmful effect on the growth and yield of pepper. In an eight-year field experiment carried out to investigate the influence of prolonged utilization of TW irrigation and fertilization practice on plant and soil properties, [201] reported that the trees irrigated with TW without fertilization practice were not adversely affected, meaning TW offers sufficient nutrients for plants. Additionally, the yield was higher in TW treatments compared to the freshwater treatment. The outcomes of seed germination and plant growth experiments proved that the TW had no negative impact on the germination responses of the seed crops (lettuce and beets). However, recalcitrant residual compounds in that water caused early stunted growth in plant root systems with limited access to available nutrients. Hence, vegetative growth and chlorophyll production were reduced [296]. Ref. [297] stated that there was a negative effect on vitamin C, protein, and the organic acid content of tomato crops that were irrigated with TW. Ref. [294] detected that there was a small increment in nitrate concentrations in tomato fruit and cucumber, but the values are still on the safe side [298]. Ref. [192] showed that tertiary TW had an affirmative impact on the yield and growth of young grapevines, while secondary TW had an adverse effect on fruit safety compared with freshwater. The quality of crops irrigated with TW is always a great concern. Ref. [261] reported no obvious HM accumulation in wheat grain that received TW. However, [279,299] reported a higher content of Cd and Pb in the maize and higher Cd content in wheat irrigated with TW than freshwater. Ref. [300] refer that the prolonged usage of wastewater may cause an accumulation of HM and plant

nutrients with unfavorable levels in the plants, thereby decreasing their quality. Ref. [301] found considerable relations between soil contents of the macronutrients, micronutrients, and HM and their accumulation in shoots and grains of maize. They reported that wheat plant contents of Pb, Cd, Zn, and Ni rose as their concentrations rose in the irrigation water. HM in non-safe limits are toxic for plants, and may result in yield reduction, and may even be accompanied by a decrease nutrient uptake, and may decrease the capability to fixate molecular nitrogen [302,303]. An experiment carried out to evaluate the effect of different HM on the vegetables irrigated with TW through SSDI, [17] illustrated that the target hazard quotient of Fe, Cu, and Zn < 1.0 appears relatively safe in all the tested vegetables. They also stated that the Health risk index (HRI) values showed that HM were less than 1.0 and, thus, less risk to humans. The HRI represents the harmful effect of heavy metals on people who consume vegetables contaminated with them [304]. When the HRI value is <1, people will be safe to eat those vegetables [305]. In the field experiment that used TW after secondary treatment to irrigate tomatoes, [268] demonstrate that HM content in the soil and crops is far lower than the national soil environmental quality standard and food hygiene permission value standards in China. Thus, usage of wastewater treated with secondary treatment would not cause a pollution to the soil environment and crops. Refs. [306,307] mentioned that the wastewater irrigation duration did not significantly affect the HM transfer factors in soil-plant systems, suggesting that HM in wastewater-irrigated soil were not easily transferred into the plant chain and likely did not cause contamination risks. Additionally, [307] mentioned that the crop grains' HM contents in wastewater-irrigated regions did not exceed allowed limits. In field experiments, [297] and [293] reported no obvious heavy metals accumulation in tomato and cabbage irrigated with TW. However, [308] found significant heavy metals' accumulation in other vegetables. Additionally, [309] reported that the irrigation with different wastewaters considerably rose the concentrations of the tested elements of Pb, Ni, Zn, Co, Cu, Cd, Mn, and Fe in vegetables, especially the leafy species. Ref. [310] studied the effects of TW on yield and the quality of cucumber and carrot. They concluded that the macro- and micronutrients in stem, leaf, and fruit were increased in plants that received TW irrigation compared to freshwater irrigation. Industrial wastewater and wastewater from slaughterhouses showed phytotoxicity to lettuce and cucumber seeds [311,312], mainly due to the concentration of heavy metal ions. Heavy metals have many direct and indirect damaging effects on plant growth [313,314]. The increasing levels of metals into the environment drastically affect plant growth and metabolism, ultimately leading to severe losses in crop yields [315]. Nickel belongs to essential elements as it has a high biological activity and toxicity [316]. Cd and Pb are recognized as the most toxic metal ions due to their detrimental effects not only on plants but also on humans, as they display the most profound mobility in the soil environment [317]. It is well documented that excess Cd or Pb inhibit plant growth, and directly or indirectly interferes with their physiological processes through disrupting their metabolism [318,319]. The heavy metals accumulate more in leafy or root vegetables than fruit organs and thus are less subject to trace elements' accumulation [3,320–322]. A major part of HM is taken up by crops from the soil via roots. HM transportation from the soil layers to the crop roots mainly depends on the soil-forming rocks type, OM content, soil pH, sorption capacity, amount of $CaCO_3$ mineral oxides, anthropogenic load, and other chemical and physical properties of the soil [323]. Globally, various field experiments have been carried out to estimate the possible microbial pollution risk of vegetables that were irrigated with TW. Many experiments have stated that the TW irrigation of the tomato is safe from the microbial aspect [324–326]. Ref. [327] demonstrated that when the broccoli and tomatoes were irrigated with food-industry TW, the outcomes revealed that no contamination with pathogenic bacteria was detected when examining the microbial content of soil and water, and no pathogens were noticed in any plant edible part. Ref. [191] demonstrated that although the soil microbial quality was affected by applied-TW irrigation, a relatively low concentration of *E. coli* was detected in soil, but no *E. coli* was observed in harvested onion, lettuce, or maize. Ref. [328] reported that fecal and total

coliforms were not presented in the grapevines irrigated with TW. A two-year study on the microbiological quality of bananas irrigated with TW and freshwater accompanied by chemical fertilizers did not show any significant differences [329]. A similar conclusion was also found with tomatoes and broccoli when irrigated with TW [16]. When lettuce was irrigated with TW using SDI, [330] stated that there are no considerable changes in soil, and there are no pathogens on the leaves of lettuce. However, [331] concluded that lettuce plants could be contaminated by *E. coli* present in the irrigation water. In an experiment carried out to estimate the effect of TW irrigation on produce safety, [332] reported that the outcomes demonstrated that the microbial pollution on the tomatoes' surface was not correlated with the irrigation waters' source; also, the bacterial pollution on the tomatoes irrigated with TW was not statistically different from the tomatoes irrigated with tap water. However, Cryptosporidium pathogens were detected in TW and on the tomato surface irrigated with this water. In a greenhouse study, there were no differences in bacterial indicators between the tap-water-irrigated crops or the TW-irrigated crops for both washed and unwashed samples [333]. However, in a field trial, they mentioned that total coliform counts were higher for all vegetables grown using TW in comparison to tap water. This was probably caused by increased contact with the soil contaminated with coliform bacteria. The observed variations between crops when irrigated with TW could be aligned to the wastewater characteristics, crop type and plant species, plant ability to strive in nutrients-deficit environments, and plant sensitivity to environmental and climatic conditions [296]. Therefore, more experiments are required to study the benefits and restrictions of reusing TW in irrigations of the crop. Ref. [17] showed that when irrigating the vegetables with TW, the proper selection of suitable vegetables and consumed parts could reduce the health risks to humans. The appropriate management of wastewater by selecting appropriate crops and irrigation management strategies, and monitoring the concentrations and the distribution of metallic trace elements in the soils and plants, can help minimize the risk of wastewater use for agriculture [17,272].

## 7. Effect of Wastewater Irrigation on Public Health

The pathogens are considered the greatest health concern and a threat to a soil's ecosystem when using TW for irrigation [334,335]. Most pathogens are eliminated or deactivated through appropriate treatment and wastewater disinfection. However, some pathogens' concentrations in the TW may still be high compared with their dose that can cause infection [336]. Public health aspects should be considered, especially if this water contains HM and OM [337–341]. Biodegradable OM included in reclaimed wastewater can raise the soil OM content, and that is considered a significant factor affecting pathogen survival [342,343]. The organic chemicals that remain in the treated water after the treatment process may contaminate and adversely affect the soil and may cause threats to the health of humans [8,111,344,345]. Many authors reported that some contaminants of emerging concern persist in reclaimed wastewater and can reach agricultural fields through irrigation, leading to their accumulation in soil [346,347] and roots [348] and translocate them to edible plant organs such as fruits and leaves [61,349–351]. For example, [352] reported that lamotrigine, carbamazepine, caffeine, sulfamethoxazole, metoprolol, and sildenafil were all persistent in soils. If pathogens are not completely abolished in conventional wastewater treatment plants, crops can take up wastewater-borne micro-contaminants [61,349]. These contaminants may accumulate in the vegetables and fruits and, consequently, reach the food chain, with major potential effects on animals and humans [8]. Moreover, these contaminants lead to contamination of groundwater [353–355]. So, to reduce these chemical risks when using treated water in irrigation, certain guidelines need to be taken into consideration. Depending on the quality of TW that directly depends on the treatments carried out [356–359], TW may carry pathogens that cause threats to humans' health [360,361]. Furthermore, this can raise the hazard for viral, parasitic, and bacterial infections in consumers of crops irrigated with this water [42,43,362–364]. Consequently, humans and organisms that may be exposed to these contaminants require

a careful periodical assessment. Many factors possibly influence the microbial load of crops and soil that received wastewater irrigation and consequently cause health risks. Ambient temperature and humidity, soil water content and pH, rate of ultraviolet radiation, antagonism with indigenous soil microorganisms, UV radiation, plant type, and method of irrigation could affect the fate and population of microorganisms in the soil and on the surfaces of crops [8,42,191,365]. Accordingly, it is important to track the fate and population of pathogenic microorganisms through experiments to assess the human health hazards related to reused TW in irrigation [191]. Therefore, many studies were carried out to study the influence of TW irrigation on the product's safety and the microbial characteristics of the soil. These studies indicated that using this water in irrigation has no significant microbial effect on the crops, and there is no microbial hazard for the consumers or the environment [326,330,332,366–372]. Additionally, [136] mentioned that the evidence about the disease spreading through irrigation with TW is scarce. However, [121,373,374] stated that bacteria, protozoa, viruses, and helminths are still present in TW after conventional wastewater treatment. Furthermore, [192] reported that soil irrigated with secondary TW was highly contaminated by total coliforms and *E. coli.* The contrast in results may be due to the difference in the wastewater characteristics and the management method. So, in all instances, to prevent disease transmission to humans from TW irrigation and ensure public health protection, the appropriate management should be followed to reduce the risks of organic micro-contaminants [111]. Ref. [191] demonstrated that through the application of in-field and postharvest control measures for reducing microorganisms and using SDI and SSDI and disinfection, washing, and peeling of produce eaten raw, public health protection could be obtained and reduce possible risks. Besides, a combination of active measures at the source, good agricultural practices such as developing irrigation methods such as SSDI, and additional preventive measures are necessary to bring about and ensure safe water reuse for irrigation.

## 8. Treated Wastewater Irrigation Systems

When using TW irrigation, selecting a suitable irrigation system that gives a uniform application and high efficiency depends on many factors such as the wastewater quality, soil and crops type, farmer's ability to manage the different methods, and the potential risk to the environment and human health. Highly productive and good yields can be achieved with TW irrigation if proper management with modern irrigation systems is applied [375,376]. Ref. [377] reported that irrigation methods could be grouped into two general types, both of which are utilized for irrigation with TW: (1) surface or gravity irrigation (e.g., furrow and border irrigation) and (2) pressurized irrigation. Pressurized irrigation can further be classified into localized irrigation (e.g., drip and bubbler) and non-localized irrigation (e.g., Gun sprinkler and center pivot). The ultimate localized irrigation methods consist of low amounts in micro-irrigation systems such as drip and micro-sprinkler. However, we must bear in mind that the large amounts of OM and suspended solids that might be present in the wastewater may cause clogging problems in system parts of micro-irrigation such as drippers and sprinklers [73,135,378]. Additionally, [208] stated that it is probable that moderately to severe chemical clogging or biological clogging will occur when using wastewater irrigation. Ref. [379] mentioned that the following general selection criteria should be taken into account when the sewage effluent is applied: (a) economic consideration, (b) topography and soil physical characteristics, (c) type of crop, (d) availability of skilled labor, (e) water quality, and (f) farming traditions. They also indicated that the selection of irrigation method and technology involves the following considerations that deal specifically with implications from TW characteristics: 1. irrigation efficiency, 2. application frequency, 3. application flow rate, 4. soil wetting profile, 5. soil salt profile, 6. foliar/fruit wetting, 7. runoff, 8. clogging hazard (emitters, pipes, filters), 9. hardware corrosion hazard, 10. de-nitrification effect, 11. offensive odor production, 12. automation, 13. filter facilities, 14. chemical and fustigation facilities, 15. drainage facilities, 16. flushing facilities for removal of clogging agents, 17. runoff and flush-water

disposal or recovery facilities, 18. facilities to include alternative sources of water, and 19. operational storage facilities.

### 8.1. Surface or Gravity Irrigation Systems

The basic principle of surface or gravity irrigation systems is that water is applied to the soil and is allowed to spread above all the soil by gravity; then, the water infiltrates into the soil and spreads over the field. The furrow and border are the widest methods used in gravity irrigation systems [380]. The design of surface irrigation systems is largely based on experience and direct field studies. In this method, it is necessary to estimate seepage losses from open and unlined channels to assess conveyance losses, and compute the required flow capacity of the delivery system, besides evaluate seepage into the groundwater [381]. It is inadvisable to apply these methods to shallow soils or soil with a high water table, particularly if the TW is of poor quality [382]. Additionally, very fine-textured soils with low infiltration rates are not appropriate for any surface irrigation method because the water flow rates in these methods are generally high. Therefore, these methods should be avoided on easily eroding soils [383].

### 8.2. Pressurized Irrigation Systems

8.2.1. Sprinkler Irrigation System

Sprinkler systems are rainfall simulators. In the sprinkler irrigation systems, the pressure head of the water is converted into a velocity head. The jet ejected from the nozzle breaks into small droplets that fall over an area where the size of which depends mainly on the operating pressure and type of sprinkler [384]. Sprinklers vary from low-flow-rate sprinklers placed at a desired spacing along a lateral to giant or gun sprinklers mounted on carriages which facilitate rapid coverage of large areas [385].

Rotating impact sprinklers with single or double nozzles are the most commonly used of all commercially available sprinklers. These sprinklers are appropriate for a wide range of operating pressures, discharges, and application rates, and practically all crops and soils. The spacing of sprinklers on the laterals and spacing of the take-off valves on the mainline are designed so that the water distribution from the sprinklers gives nearly complete overlap [386]. For TW irrigation, mechanically moving systems, in which the lateral line is carried on wheels, will reduce labor-health hazards because of reducing the physical contact between the TW and laborers, compared with moving these laterals by hand [387]. The sprinklers used with TW irrigation should be selected and designed to reduce aerosols' drift, which represents a probable health threat. So, it is recommended to select sprinklers with low-pressure nozzles.

Solid-set sprinkler systems are considered the most suitable for projects of effluent disposal at high-application rates. These systems may also be equipped with gun sprinklers. The high initial cost of these systems is compensated by decreasing labor need and increasing efficiency [388]. It is also recommended to irrigate during low-wind-speed periods and at night when evaporation is low. Additionally, it is recommended to make buffer zones around the site [389].

Center-pivot systems include a lateral pipeline, generally made of galvanized steel, anchored to a fixed central pivot structure around which it continuously rotates and covers a circular area. Water is supplied to the lateral pipe through the pivot. The lateral pipe with the desired sprinklers is supported on 7–10 self-propelled towers on wheels at about 30–60 m apart. The sprinklers are arranged on the lateral line to produce uniform water distribution while making a complete revolution. The sprinkler can be replaced by spray nozzles on vertically suspended flexible tubes close to the crop to reduce aerosol drift and apply the effluent near the soil [387].

Travelling lateral or side-roll systems that move continuously in a rectilinear fashion are available and are becoming increasingly popular for row crops. In these systems, sprinklers can be replaced by trickle emitters that apply the water on the ground, thus

eliminating aerosol drift. Additionally, in these systems, the lateral line can be fed either from a flexible drop hose or by a travelling pump unit that pumps the water from a ditch.

8.2.2. Drip Irrigation System

The drip irrigation system is a technique in which the irrigation water flows out of a filter into special drip pipes, with emitters located at different spacing, and the irrigation water is distributed out of these emitters directly to the ground near the roots [112]. If the drip irrigation system is properly designed, installed, and managed, it may help achieve water conservation by reducing evaporation and deep drainage. Compared with other irrigation systems such as flood irrigation or overhead sprinklers, irrigation water can be precisely applied to the plant roots. The small quantity of water reduces weed and grass growth and limits the leaching of plant nutrients down in the soil [390]. Additionally, drip irrigation can eliminate many diseases that are spread through irrigation water. Drip irrigation is adaptable to any farmable slope and is suitable for most soils. In contrary to commercial drip irrigation, simple self-made systems are cheap and effective [391]. The drip-irrigation method is safe and generally is the most suitable irrigation method when using wastewater. In this method, the physical contact between the wastewater and both the crops and the farmers is minimized. Besides, there are no aerosols in this method and, thereby, no atmospheric pollution can occur. Drip irrigation is the most efficient irrigation application for saline water [141,392–394]. In a field study on tomatoes, [332] indicated that under SDI with TW, the fecal bacteria or microbial pathogens did not transfer to the irrigated soil or crop. In the study conducted by [282], the results revealed that SSDI offered more reliable growth and yield data than SDI. SSDI has proven to be an efficient irrigation method to provide the best health protection [391]. It can effectively protect farmers and consumers by minimizing crop and human exposure to irrigation water [320]. The application of secondary TW under SDI and SSDI systems affects the survival of soil microorganisms, thus reducing the environmental risk [395]. In comparison with SDI, the SSDI appeared to be effective in declining pathogens' number in irrigation water and limiting their presence on the soil's surface [396]. In a study conducted to study the impact of using TW with different irrigation systems on tomato, [397] suggested that the application of SDI and SSDI can control the environmental contamination and decline the soil pollution problems. Additionally, maximum yield gained under drip irrigation treatments compared with other irrigation systems, and that may be as a result of the better soil moisture and raised content of the available nitrogen in the root zone. Moreover, they reported that among wastewater treatments, the SSDI had the better tomato microbiological quality. Ref. [398] reported that no contamination was noticed in the TW irrigation treatments under SSDI, but *E. coli* and fecal coliforms were noticed in the surface soil samples of TW irrigation treatments under SDI. In a study carried out to investigate the incidence of *E. coli* in bitter gourd, cauliflower, and soil profiles irrigated with wastewater under SDI and SSDI, the results indicated that SSDI treatments had the minimum concentration of *E. coli* for cauliflower and bitter gourd compared with SDI [399]. Comparing sprinkler irrigation with the SSDI with TW, using an irrigation depth of 100% of the daily evapotranspiration allowed proper development of the zoysiagrass lawn by maintaining its quality, with no contamination by *E. coli* or total coliforms [400]. The drip-irrigation system can reduce the load of total coliforms and *E. coli* counts. SSDI has a critical role in decreasing the load of coliform in the soil and the crops, ensuring the safety of the consumers against health hazards [401]. In an experiment conducted in a greenhouse in Tunisia to assess the influence of different irrigation methods with TW on the concentration of metallic trace elements in corn and soil, [272] stated that the lowest levels of metallic trace elements and salinity were recorded with using SSDI. Furthermore, the largest amounts of nutrients elements were found under SSDI. SDI and SSDI methods are the most suitable irrigation methods to be used with TW because they reduce crop contamination, and mitigate human health risks by reducing direct contact between TW and plant [402].

## 9. Conclusions

Reused TW irrigation has become a valuable resource and an attractive option to manipulate water lack, especially since the wastewater amount is huge in several countries. Using this huge wastewater as an irrigation source after appropriate treatment has economic and environmental benefits since it could conserve a huge quantity of freshwater, besides decrease or even eliminate the need to supply expensive chemical fertilizers to the soil. There are inconsistent results on the influence of TW irrigation on crops, which might be because of the characteristics of wastewater, crop type, plant species, the plant's ability to strive in a nutrient-deficit environment, and plant sensitivity to environmental and climatic conditions. There is a possibility of negative effects of prolonged irrigation with TW on the deterioration of soils' physicochemical properties and increased soil microbial activity. Therefore, proper guidelines for wastewater reuse and management should be followed to limit any negative effects. More studies are needed to study the limitations and benefits of reusing TW in crop irrigation. SSDI is the most suitable irrigation technique for using wastewater because it has proven to be effective in declining pathogens' number in irrigation water and limiting their presence on the soil's surface. Besides, it reduces crop contamination and mitigates human health risks by reducing the physical contact between the wastewater and both the crops and the farmers.

**Author Contributions:** M.S.H. and X.Q.; methodology, M.S.H. and X.Q.; investigation, M.S.H. and X.Q.; resources, M.S.H. and X.Q.; data curation, M.S.H. and X.Q.; writing—original draft preparation, M.S.H. and X.Q.; writing—review and editing, M.S.H. and X.Q.; visualization, M.S.H. and X.Q.; supervision, X.Q.; project administration, X.Q.; funding acquisition, X.Q. All authors have read and agreed to the published version of the manuscript.

**Funding:** This review paper is supported by the National Natural Science Foundation of China (grant No. 51679241, 51709265) and the Agricultural Science and Technology Innovation Program (Grant No. CAAS-ASTIPFIRI-03).

**Institutional Review Board Statement:** Not applicable.

**Informed Consent Statement:** Not applicable.

**Data Availability Statement:** Not applicable.

**Conflicts of Interest:** The authors declare no conflict of interest. The funders had no role in the design of the study; in the collection, analyses, or interpretation of data; in the writing of the manuscript; or in the decision to publish the results.

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
