# Peer review of "Treated Wastewater Irrigation—A Review"

_water, doi:10.3390/w13111527_

Round 1

Reviewer 1 Report

A thorough English revision is mandatory. The phrase constructs in the vast majority of the document are poor, and the sentences verbe tense is, quite often, inappropriate. A significant fraction of the document, as it is written, lacks sense. A large number of typos, inappropriate use of capital letters and misused terms should also be corrected. Given the quite poor English quality of the document (practically all sentences in the document. in its present form, need to be corrected), proof editing by a native English speaker is highly advisable.

A number of fundamental chemical, biological, wastewater and WWT notions are not being backed up by the appropriate literature (such as reference books in this field of study) but solely by magazine articles. This should be corrected. Furthermore, the references section also needs a thorough revision given the large number of incomplete or incorrectly stated references.

The presented literature is not fully integrated, and a more heuristic perspective would be advantageous. In a number of studies the authors come to a given conclusion opposite to other sets of studies. Both types of studies should be better confronted in order to derive a unifying conclusion. This was not fully performed.

The authors should also address the following issues.

The treated wastewater (TW) definition, presented in line 56, should be presented earlier in the document.

In lines 76-78 the authors state “The secondary treatment process aims to diminish the BOD exerted by removal or reducing of residual settleable solids and floating materials from primary treatment”. The secondary treatment is much more comprehensive than solely the "removal or reducing of residual settleable solids" aiming to dimmish not only BOD, but also COD, SS and a set of other parameters (TN, TP, etc).

In lines 79-80, the authors state “Also, 80 decreasing the microbiological constituents by using chlorine in disinfection operation.” There a significant number of other options that can be used, rather than chlorination, such as UV, ozonation, among others.

In lines 81-82, what do the authors mean by “There are no significant constraints on the utilization of the secondary TW as a fertigation source [38].”? Do the authors mean that TW may be employed prior to biological stabilization?

In line 108 the sentence “Product of complex nitrogenous OM. [55]–[58].” should be a part of the previous sentence and not appear on its own.

In lines 113-115, the authors state “But [63] demonstrated that these particles are separated by sludge settling and skimming of solid particles in primary wastewater treatment.” However, sludge settling of microplastics in primary WWT is far from being able to fully remove microplastics from WW. The authors should correct this information. For instance, according to some reports*, 80% of the MP in natural waters is derived from WWTP.

* Raju et al. (2018). Transport and fate of microplastics in wastewater treatment plants: implications to environmental health. Reviews in Environmental Science and Bio/Technology 17(4), 637-653. https://doi.org/10.1007/s11157-018-9480-3

In lines 119-120, the authors state “Also, they mentioned that the ecotoxicological hazard of pharmaceutically compounds in the soil was very low.” However, this is widely dependent on the PhAC as shown by Leal et al. (2020). Environmental impact and biological removal processes of pharmaceutically active compounds: The particular case of sulfonamides, anticonvulsants and steroid estrogens, Critical Reviews in Environmental Science and Technology, 50:7, 698-742, DOI: 10.1080/10643389.2019.1642831

In table 1, “PH” should be replaced by “Acidity”, and Eubacteria and Archeobacteria terms are no longer in use. The authors should replace them by Bacteria and Archae.

In table 2, “Communicable diseases” should be placed by “Infectious diseases”.

In lines 151-152, why are bacteria singled out from the aquatic life found in water?

In lines 153-154, why is not mentioned that a number of microalgae present in wastewater (or in wastewater treatment) are capable of producing powerful toxic substances, harmful to animals (including humans)?

In lines 160-162, the authors state “Biological wastewater characteristics can be derived with the help of measuring both Chemical Oxygen Demands (COD) and Biological Oxygen Demand (BOD) [70].” Can the authors elaborate more?

In lines 165-166, please replace “volume of dissolved oxygen” by “amount of dissolved oxygen”. In a number of other instances throughout the document “volume” should also be replaced by “amount”.

In line 177, please replace “organic compounds' numbers” by “organic compounds' contents”.

In line 181-182, the authors state “A high COD/BOD ratio could consider a pointer of the toxins existence.” Please explain further (and correct the grammar structure).

In line 182, the authors state “The (COD/BOD) ratio for sewage is usually about 2:1 [76].”. However, this ratio is extremely dependent on the wastewater.

Lines 214-217 repeat lines 211-214. The same happens in lines 475-478 and 933-935.

In table 4, please introduce the acronym TDS (Total Dissolved Solids?).

Lines 306-310 do not provide real useful information for the reader. Please substitute the topic like structure of this sentence by adequate information. For instance, instead of "select irrigation system" present the irrigation systems that should be used to reduce the salinity effect, etc.

In line 311, should it really be “Alkalinity” the title of this section?

In lines 347-350, the authors state “Heavy metals that exist in municipal wastewater (such as Arsenic (As), copper (Cu), Cadmium (Cd), nickel (Ni), molybdenum (Mo), Chromium (Cr), zinc (Zn), and Lead (Pb)) are effectively removed by Conventional treatment processes until their concentrations are near to the concentrations in freshwater [99], [159]–[162].” This sentence is debatable regarding most widely used conventional wastewater treatment systems (such as activated sludge systems). I advise the authors to rephrase this sentence.

Why is the sentence in lines 366-370 presented within the heavy metals section, when it refers to the nitrate and nitrite concentrations?

The sentence in lines 433-435 seems incomplete.

In lines 467-468 ESP is erroneously described as cation exchange capacity. Please correct it.

In lines 496-498 please elaborate more on the soil microbial component and soil enzyme activities role in the environment of soils.

In lines 553-554, the authors state “However, the high temperature helps in the reduction and elimination of microbial risks [243], [244].” What do the authors mean by "high temperature"? Prior elimination of microorganisms by heat treatment in the TW?

In line 569, what do the authors mean by “soil solution”?

In line 597, the heavy metals concentrations in the field irrigated with TW were found to be slightly higher or slightly lower than in the field irrigated with freshwater?

In line 610 and line 699, please replace “HM volumes” by “HM contents”.

In lines 625-627, the authors state “In China, the cereal crops irrigation with TW has increased rapidly since 2002 to maintain the water shortage [96].” This sentence lacks sense to me. Irrigation with TW is used to prevent water shortage as opposed to maintain water shortage.

In line 642, what do the authors mean by “yield attitudes”?

The sentence in lines 642-643 seems incomplete.

In lines 696-698, please elaborate more on the risk index and why values less than 1 are considered safe.

In lines 753-757, the authors state “However, in a field trial, they mentioned that total coliform counts were higher for all vegetables grown using TW in comparison to tap water. This was probably caused by increased contact with the soil contaminated with coliform bacteria. Although, fecal coliform and E. coli were not found in any samples of vegetable grown in this field trial [329].” The authors start to state that the coliform count in a field trial was higher for the use of TW and finish stating that no E. coli and fecal coliform were found in the field trial. Either the first part or the second part is true, it cannot be both true.

In line 764, what do the authors mean by “appropriate management”?

Section 7 should be rearranged. The authors do not follow a fully coherent structure within this section, jumping from one subject to the other and returning to older subjects without an entirely logical construct.

In line 782, what do the authors mean by “wastewater-borne micro-contaminants”?

In lines 8058-809, the authors state “Conventional wastewater treatment plants reduce E. coli, other coliform bacteria, and intestinal enterococci [368].” I believe this sentence not to be entirely true, with respect to systems not comprising a disinfection unit.

In line 856, what do the authors mean by “showily soils”?

In line 915, what do the authors mean by “saline water”?

Other minor issues needing correction refer to the following. Liter should be represented by L. The authors should avoid the use of /x and use x-1 instead. In a variety of occasions pH appears as PH in the document. This should be corrected. Also, please use the full species name the first time it is mentioned in the document (for instance, Escherichia coli), and the abbreviation in later mentions (for instance, E. coli).

Reviewer 2 Report

The review concerns the issue of the  treated wastewater irrigation impacts on soil properties (biological, chemical, and physical), and fertility status.

Remarks: The title should be corrected, as it does not reflect the content of the paper, as it is too general. In reference abbreviation of journal papers and DOI number should be presented. Authors should improve the presented references according to the journal’s guidelines. In what way this review will help the research community, policymakers, and other stakeholders? To clarify some aspects, I would suggest that the authors write the bibliography evenly: the abbreviation, the DOI number. In what way the publications were chosen to perform the analysis? Are they indexed in Wos or Scopus? Also, quite old works were taken into account. Weak conclusions. The abstract should be more concise. There is a lack of careful analysis of the results of many experiences of cited papers. Articles do not indicate detailed discrepancies in published research results, so a lack of synthesis of conclusions resulting from the survey review occurs.

Reviewer 3 Report

Type pf article:  It is not “Article” it is a review paper. It should be changed.

Abstract: well written and organised, the main objective of study should clearly underlined

Key words: numbering should be removed

Introduction:

Please improve the style of citing, as “s [1]–[3]”, and “[4], [5]”  is not appropriate, should be [1-3]. Please revise in all text.

All shortcuts should be explained in the text when they are used for the first time, as “TW”, “BOD”, in Introduction. Please revise all text.

Page 2, line 44, “of TW in these countries is encouraged”. Which countries? It should be clarifies. Before the regions were mentioned, without indication which specific regions. Please add some examples as well.

Page 2, line 45, “china’ should be replaced by “China”

Please improve the units in all manuscript, including “m3” (should be m3)… For China authors used “m3” and for Egypt „cubic metres”. It should be harmonised in all text of manuscript.

There is some mix of  used tenses, please revise the whole text.

Page 2, lines 70-79, Make a point in the following lines, as:

1)….

2)…

The Introduction should be improved.  The main objective of the study should be clearly presented, supported by the sub-objectives and scope of the research.

Authors indicated that “The aim of this article review is to reviews the treated wastewater irrigation impacts on soil properties (biological, chemical, and physical), and fertility status..” Before, they described water/wastewater reuse importance. What is the relevance to soil quality. In my opinion, the described background is more relevant to article about water reuse, than the treated wastewater irrigation impacts on soil properties. The context of the conducted research should be more underlined, and importance to field of science.

Please provide the novelty of the work.

This Introduction should be rewritten, according to the structure which should be included in the research papers. The introduction of a research paper should contain a few other parts/ elements such as the chief goal(s) and objectives of the research, a brief but informative outline of the following content, explained, concept definitions, a brief history of the research into the topic, recent related discoveries, etc.).

Materials and methods: There is lack of this section in the work. Please provide step-by-step procedure of research, including description of used materials and methods.

  1. Characteristics of wastewater:

What is the context of this section? It is not clear from the description of the Introduction why authors revised the wastewater quality.

What is the added value of the Tables 1 & 2? As it is indicated as direct copy-paste from one reference [70], it does not provide any new knowledge proposed by authors. The same for Table 3 with reference [98]. Usually in the review papers, the Tables proposed by the Authors are the integration of knowledge from many authors (papers). The direct copy-paste from other Authors it not recommended.

The same comment for other Tables.

Conclusions:

Please provide the conclusions, not repetition of the results.

References:

Should be adapted to journal template.

Overall:

There is some mix of  used tenses, please revise the whole text.

A lot of grammar errors in the text. Should be improved.

Round 2

Reviewer 1 Report

Although most of the raised concerns were appropriately dealt with by the authors, this manuscript English is now far more suitable than the previous version, yet it still needs further revision. A copy of the manuscript highlighting the major instances where it still needs correction was uploaded (words needing to be eliminated have been identified by strike by a red line and words needing to be changed are highlighted in yellow – in most highlighted cases the verb tense needs to be corrected). As referred in the prior revision, liter should be represented by L (capital letter) in all instances.

The References section must be thoroughly corrected. A number of entrances only show the first authors when it should be showing all authors; some journal articles present the publisher and city (which should not appear) and, instead, in the books and book chapters often the publisher and city do not appear (and should appear). In some instances the text should be italicized and it is not whereas in other instances the opposite occurs. Please thoroughly revise this section in accordance with this journal rules.

Author Response

Thank you very much indeed for your accurate review of the manuscript, your comments, corrections of errors, and your great contribution to improving this manuscript.

Point 1: Although most of the raised concerns were appropriately dealt with by the authors, this manuscript English is now far more suitable than the previous version, yet it still needs further revision. A copy of the manuscript highlighting the major instances where it still needs correction was uploaded (words needing to be eliminated have been identified by strike by a red line and words needing to be changed are highlighted in yellow – in most highlighted cases the verb tense needs to be corrected). As referred in the prior revision, liter should be represented by L (capital letter) in all instances.

Response 1: We downloaded the manuscript and tracked all the highlighted points you marked. Also, we have replaced the letter l, which indicates a liter to L.

Point 2: The References section must be thoroughly corrected. A number of entrances only show the first authors when it should be showing all authors; some journal articles present the publisher and city (which should not appear) and, instead, in the books and book chapters often the publisher and city do not appear (and should appear). In some instances the text should be italicized and it is not whereas in other instances the opposite occurs. Please thoroughly revise this section in accordance with this journal rules.

Response 2: We downloaded the RIS cite file of each reference from the google scholar website, then we used Mendeley (reference manager program). And we used the IEEE style. But We have improved the style of citing manually, as well as the references section.

Reviewer 2 Report

Accept in the present form.

Author Response

- Accept in the present form.

Thank you very much.

Reviewer 3 Report

Content of article was improved.

Please revise all units.

Please improve quality of Tables (position in the text and the font and its size).

Author Response

Point 1: Content of article was improved.

Response 1: Thanks. This happened thanks to your appreciated review and comments.

Point 2: Please revise all units.

Response 2: We have revised all the units.

Point 3: Please improve quality of Tables (position in the text and the font and its size).

Response 3: We have improved all tables accordingly.